# Obeticholic Acid for Primary Biliary Cholangitis

**DOI:** 10.3390/biomedicines10102464

**Published:** 2022-10-02

**Authors:** Annarosa Floreani, Daniela Gabbia, Sara De Martin

**Affiliations:** 1Department of Surgery, Oncology and Gastroenterology, University of Padova, 35131 Padova, Italy; 2Scientific Institute for Research, Hospitalization and Healthcare, 37024 Verona, Italy; 3Department of Pharmaceutical and Pharmacological Sciences, University of Padova, 35131 Padova, Italy

**Keywords:** primary biliary cholangitis, obeticholic acid, ursodeoxycholic acid, farnesoid X receptor

## Abstract

Primary biliary cholangitis (PBC) is a rare autoimmune cholestatic liver disease that may progress to fibrosis and/or cirrhosis. Treatment options are currently limited. The first-line therapy for this disease is the drug ursodeoxycholic acid (UDCA), which has been proven to normalize serum markers of liver dysfunction, halt histologic disease progression, and lead to a prolongation of transplant-free survival. However, 30–40% of patients unfortunately do not respond to this first-line therapy. Obeticholic acid (OCA) is the only registered agent for second-line treatment in UDCA-non responders. In this review, we focus on the pharmacological features of OCA, describing its mechanism of action of and its tolerability and efficacy in PBC patients. We also highlight current perspectives on future therapies for this condition.

## 1. Introduction

Primary biliary cholangitis (PBC) is a chronic disease characterized by the accumulation of bile acids in the liver, potentially progressing to cirrhosis, end-stage liver disease, hepatocellular carcinoma, and even death [1]. The existence of gender differences in PBC development has been widely reported. Indeed, PBC develops more frequently in females than males [1]. In the global population, a prevalence of 14.6 cases per 100,000 people has been observed, with a female:male ratio of 9:1, and 1.76 new cases diagnosed per 100,000 people each year [2]. Due to more careful routine testing and/or incompletely understood changes in environmental factors, the definition and outcome of PBC have been reconsidered over the last 30 years, from a severe symptomatic disease characterized by symptoms of portal hypertension to a milder disease with a long natural history [3]. As a consequence, many patients are asymptomatic, and most new diagnoses (up to 60%) are made after the discovery of increased serum biochemical markers of liver function during check-ups performed for unrelated purposes [4,5]. This autoimmune cholestatic disease is characterized by increased plasma levels of alkaline phosphatase (ALP) and the presence of a high titer of antimitochondrial antibodies (AMAs) in over 90% of patients, as well as a PBC-specific anti-nuclear antibody (ANA). The current EASL guidelines suggest that a diagnosis of PBC can be determined in adult patients in the presence of cholestasis and the absence of other systemic diseases, when the ALP value is elevated and AMAs are present with a titer >1:40 [6].

Ursodeoxycholic acid (UDCA) represents the gold standard for PBC therapy, and it is generally administered as a daily oral treatment (recommended dose: 13–15 mg/kg) [6]. UDCA therapy improves liver transplantation (LT)-free survival in PBC patients, including those with early and advanced disease, and also in patients who did not meet the accepted criteria for UDCA response [7]. Even though the improvement of biochemical parameters after UDCA treatment is modest, patients experience a long-term benefit in terms of improved survival. Regardless, non-responders represent 30–40% of all UDCA-treated patients, and globally have a higher risk of PBC progression and a greater need for transplant than responder patients, as well as a higher mortality [8]. A young age at diagnosis and male sex have been associated with a reduced chance of biochemical response to UDCA therapy in a large cohort study from the UK-PBC study group [9]. Accordingly, another large, multicenter long-term follow-up study (*n* = 4355) found that young PBC patients (aged <45) had significantly lower response rates to UDCA than their older counterparts (aged >65) [10]. However, the biological mechanisms underpinning this clinical observation in non-responders to UDCA are far from completely understood.

Therefore, the proposal of a second-line therapy devoted to UDCA non-responders provides the rationale to overcome the observed limitations of drug efficacy. To date, obeticholic acid (OCA) represents the only second-line treatment recommended for non-responder PBC patients, which are intolerant to UDCA therapy or in whom a 12 month-treatment haven’t produced benefit. As demonstrated by clinical trials, including the phase III POISE study described in detail below, OCA is effective in improving the serum and histological endpoints of PBC patients in monotherapy. In this review, we focus on the mechanism of action of OCA and its tolerability and efficacy in PBC, and offer a perspective on the future treatment of this condition.

## 2. Pharmacological Actions of OCA

OCA, a synthetic derivative of the bile acid (BA) chenodeoxycholic acid, is an agonist of the farnesoid X receptor (FXR) [11], a key nuclear receptor mainly expressed in the liver and gut, which orchestrates complex signaling pathways related to the homeostasis of bile acids (BAs) (Figure 1). In vitro pharmacological studies have demonstrated that OCA is an FXR agonist with a potency 100 times higher than endogenous BAs [12]. BA synthesis occurs in the liver starting from hepatic cholesterol. After their synthesis, BAs are secreted into the gut to help digestion and consequently the absorption of nutrients, in particular lipids and liposoluble vitamins, by virtue of their emulsifying ability [13]. After their secretion, about 95% of BAs are reabsorbed from the terminal ileum, thus entering into the enterohepatic circulation. As FXR agonists, BAs themselves participate in the finely tuned regulation of their own synthesis and secretion through the modulation of FXR activation. In PBC-related cholestasis, the enterohepatic circulation of BAs is impaired, leading to hepatic inflammation and damage.

Similar to other nuclear receptors [14,15], upon activation, FXR binds to the retinoid X receptor (RXR). The binding of the FXR–RXR heterodimer to DNA responsive elements results in the induction of the small heterodimer partner (SHP) gene, finally causing the transcriptional repression of rate-limiting enzymes in BA synthesis, such as cytochrome P450 (CYP)7A1 and liver receptor homolog 1 (LRH-1) [16]. LRH-1 is a transcription factor with a key role in the regulation of BA and cholesterol homeostasis, and also in coordinating a panel of other hepatic metabolic processes [17]. In addition, FXR stimulates the synthesis of fibroblast growth factor-19 (FGF-19), which in turn participates in the inhibition of CYP7A1 and CYP8B1 expression through the fibroblast growth factor receptor-4 (FGFR4) pathway in hepatocytes [18]. As a result, the above-described FXR/SHP and FXR/FGF19/FGFR4 pathways are major negative regulators of BA synthesis. Furthermore, FXR inhibits the sodium taurocholate co-transporting polypeptide (NTCP) via SHP, thereby repressing hepatic BA uptake [19]. FXR activation also increases the efflux of BAs from the liver to the canalicular lumen by targeting the transporter bile salt export pump (BSEP) and multidrug resistance protein-3 (MDR3), triggering another mechanism responsible for the anticholestatic effects of FXR agonists [20]. FXR activation also leads to an increase in the expression of the organic solute transporters OSTα and β, which also enhance BA efflux from the liver to the portal vein [21]. Besides its pivotal activity as a BA-responsive transcription regulator of BA synthesis and metabolism, as described in detail above, it has been demonstrated that FXR-mediated signaling plays a role in hepatic fibrogenesis, although controversial results have been obtained regarding this function. Hence, it has been observed that FXR knock-out mice develop hepatic inflammation, fibrosis, and liver tumors over time [22] and, accordingly, it has been demonstrated that OCA-induced FXR activation reduced liver fibrosis in two different experimental in vivo models of liver fibrosis [23]. Other authors have suggested that FXR in liver fibrosis models can be either detrimental or irrelevant, depending on the type of damage [24]. Notably, no direct effects of FXR agonists could be observed on the activation of cultured hepatic stellate cells (HSCs) [25,26], which are the main cell types triggering the fibrogenesis process [27].

OCA exerted both anti-inflammatory and ant-fibrotic effects by targeting the activation of both liver sinusoidal endothelial cells (LSECs) and Kupffer cells [26]. In particular, OCA reduces the production of inflammatory cytokines and chemokines (transforming growth-factor β, connective tissue growth factor, platelet-derived growth factor β-receptor, monocyte chemo-attractant protein-1) by these two types of sinusoidal cells, which in turn activate HSCs [28]. Hence, the mechanism of the anti-inflammatory effect relies on the inhibition of the NF-κB signaling pathway via the up-regulation of its inhibitor IκBα. In summary, OCA acts by a complex mechanism, comprising several actions: (a) the regulation of bile acid transport; (b) the reduction in inflammation; (c) the modulation of cellular pathways triggering fibrogenesis [29]. Due to the induction of a signaling pathway which modulates the activity of fibroblast growth factor-19 (FGF-19), OCA exerts greater hepatoprotection than UDCA. OCA also induces the expression and secretion of gut-derived hormones, e.g., FGF-19 [30]. This hormone is absorbed and secreted by enterocytes into the portal blood, thereby reaching the liver through the portal venous system. In the liver, FGF-19 is involved in the anticholestatic mechanisms described above.

## 3. Pre-Registration Studies

OCA has been evaluated in monotherapy in a phase II study in which PBC patients were enrolled with the aim of assessing its benefit in the absence of UDCA treatment [31]. After randomization, patients were treated with a placebo (23 patients), or two doses of OCA (10 mg in 20 patients and 50 mg in 16 patients) for 3 months, and followed up by a 6-year open-label extension. The ALP reduction, measured as the percentage difference from the baseline, was evaluated as the primary endpoint of this study. The treatment with both dosages induced a significant ALP reduction compared to the placebo. Accordingly, other plasma parameters were reduced in OCA-treated patients, e.g., conjugated bilirubin, GGT, AST, and immunoglobulins. In this study, the most common adverse effect reported after OCA treatment was pruritus, having been experienced by 15% of the 10 mg-treated patients and 38% of the 50 mg-treated patients.

The first approval of OCA was obtained following the results of a phase III trial that enrolled 216 patients [32], and demonstrated that about 59% of UDCA-non-responders benefitted from a one-year treatment with a combination of OCA and UDCA. These patients reached the clinical endpoint, set as an ALP level of less than 1.67 times the upper limit of the normal range, with a reduction of at least 15% from the baseline). Thereafter, the study underwent an open-label extension phase in which 193 enrolled patients were switched to OCA treatment [33]. The results of the following 3-year interim analysis showed that OCA therapy was well tolerated and could be demonstrated to maintain its performance over time. Additionally, a post-hoc analysis revealed that OCA induced a significant bilirubin reduction (both total and direct) that was particularly evident in those patients with a high baseline value of direct bilirubin [34]. This analysis thus confirmed the beneficial effects of OCA therapy in high-risk patients. Furthermore, the histological analysis of liver biopsies at baseline and after a 3-year treatment with OCA in a subgroup of patients (*n* = 17) revealed the improvement or stabilization of a panel of histologic disease features, e.g., ductular injury, fibrosis, and collagen morphometry [35]. This analysis, despite the limited number of assessed liver biopsies, further demonstrated that OCA is effective in UDCA-non-responders. The most reported adverse effects related to OCA treatment were pruritus and fatigue, which were experienced by 77% and 33% of patients, respectively [34]. As regards pruritus, only 8% of the OCA-treated patients interrupted the treatment during the open-label extension phase and, in general, patients reported a mild-to-moderate pruritus, and those experiencing severe pruritus were treated with specific medication after a clinical consult. In general, the results of this clinical trial demonstrate that 3 years of OCA treatment were efficient in ameliorating or stabilizing multiple histological features of PBC in most patients with an inadequate UDCA response, and supported the approval of OCA from the FDA in 2016.

Another sub-analysis of the above-reported trial observed that OCA treatment induced a significant reduction in the AST to platelet ratio (APRI). This effect was observed after a 1-year treatment and in the open-label extension phase in the groups treated with 10 and 50 mg OCA with respect to the placebo [36]. Liver stiffness (LS) was evaluated in 39 patients randomized and dosed with the placebo, 35 patients dosed with OCA 5–10 mg, and 32 patients dosed with OCA 10 mg. LS at baseline was 12.7 ± 10.7, 10.7 ± 8.6, and 11.4 ± 8.2 kPa, respectively. During the double-blind and open-label phases, a decrease, while not significant, was only observed in the OCA 10 mg group, while both the OCA 5–10 mg and placebo groups displayed mean increases in liver stiffness [36]. In other words, a trend towards a reduction in LS was observed only in the arm treated with the highest dose of OCA. In another scenario, namely non-alcoholic steatohepatitis, patients enrolled in the phase III REGENERATE study with OCA showed a significant reduction in LS after 18 months in the OCA 25 mg group vs. the placebo [37]. Thus, the assessment of the antifibrotic activity of OCA in a clinical setting has several limitations, mainly considering that changes in LS occur during a median interval of 2 years.

The main pre-registration studies evaluating the efficacy and safety of OCA are reported in Table 1.

## 4. Real-World Data on OCA

Currently, OCA is available as tablets containing 5 and 10 mg under the brand name Ocaliva. Typically, therapy for PBC patients is started with the administration of an initial dose of 5 mg once daily, which can be titrated to a maximum of 10 mg daily [40]. The general recommendation for patients with advanced cirrhosis (Child–Pugh B or C) is to start with a dose of 5 mg once weekly, which is then increased to a maximum of 10 mg twice weekly if the drug is well-tolerated.

The most significant ADRs caused by OCA therapy which have been reported in clinical trials are pruritus, fatigue, nausea, and headache. To a minor extent, hypersensitivity reactions and depression have also been observed [40]. As far as pruritus is concerned, it appears to be less severe if the patients are initially treated with a low dose, which can then be gradually increased. As a consequence of the alteration of lipid metabolism, which is due to other molecular signaling pathways triggered by FXR activation, an increase in total serum lipid levels and a small decrease in high-density lipoprotein (HDL) have also been reported in PBC patients treated with OCA, but to date these effects have not been correlated to a long-term increased cardiovascular risk [30].

Real-world data are crucial for understanding treatment effectiveness and safety in everyday clinical practice where: (i) patients’ characteristics are more heterogeneous with respect to sub-phenotypes, e.g., cirrhosis and overlap syndrome between PBC and AIH; (ii) the treatment schedule may be less rigid and more “personalized” by each treating physician. A number of post-registration clinical trials are ongoing and recruiting patients (Table 2).

Three real-world cohorts have been published thus far (Table 3), all reporting results for 12 months of OCA treatment [41,42,43]. Altogether, 375 patients treated with OCA were included in these three studies. The main characteristics of the three cohorts are respectively described in Table 3. The inclusion criteria were: hepatologist’s discretion for the Canadian cohort, lack of response to Paris II criteria [44] for the Iberian cohort and ALP >1.5 times the normal according to the Italian Medicines Agency (AIFA) for the Italian cohort. The percentages of patients with cirrhosis were 6.3, 10, and 15%. The percentages of response at 12 months according to the POISE criteria were respectively 18, 29.5, and 51.9%. Due to the retrospective design of these studies, a comparable evaluation of the response to OCA is impossible. However, it has to be pointed out that in the Italian cohort, with one third of cirrhotic patients, the response rate was lower due to the higher drop-out and higher levels of bilirubin at baseline in cirrhotic patients. Within the Canadian cohort, 11 patients (17%) had a permanent discontinuation of treatment (2 of them with Child–Pugh A and B respectively) for suspected hepatotoxicity. The first case was a 67-year-old female who discontinued OCA due to an increase in ALP. The second patient was a 54-year-old female who developed severe cholestatic cirrhosis, who was transplanted for severe complications. Within the Iberian cohort, a total of 14 patients (11.67%) discontinued the treatment due to severe adverse events or decompensation of cirrhosis. Within the Italian cohort, 33 patients (17%) discontinued OCA for pruritus or other side-effects. In the same cohort, factors associated with a lack of response at 12 months were: previous treatment with fibrates, high levels of ALP at baseline, and high levels of bilirubin at baseline [43].

A further analysis was performed in 100 cirrhotic patients of the Italian cohort [45]. The response to treatment was obtained in 41% of cases, according to the POISE criteria, confirming OCA efficacy at this stage as well. In this case, the use of the normal range criteria means that the endpoint was reached by only 11.5% of the cirrhotic patients. Regarding the reported severe adverse effects, 22% of patients discontinued OCA therapy: 5 patients due to jaundice and/or ascitic decompensation, 4 due to upper digestive bleeding, and 1 subject died after the substitution of a transjugular intrahepatic portosystemic shunt.

A sub-analysis from the Italian and Iberian cohorts found that patients with PBC/AIH overlap syndrome had a similar response after OCA treatment [42,43].

Two further real-world studies were presented at an AASLD virtual meeting in 2020. The first study, derived from the GLOBAL PBC group, enrolled 290 patients in 11 centers located between Europe, North America, and Israel [46]. Among them, 215 patients met the POISE criteria for eligibility, 60 patients possessed available biochemical data for a period of 12 months, and 35% of patients reached the pre-defined POISE primary endpoint after 1 year of treatment. The second study was conducted on 319 patients that received OCA therapy between May 2016 and September 2019, and were considered eligible for OCA according to laboratory databases and American administrative claims [47]. According to the Toronto criteria, the proportion of patients achieving a biochemical response to the treatment was 48% after 1 year, 58% after 2 years, and 55% after 3 years which marked the end of the follow-up period [48]. More recently, a large nationwide experience of second-line therapy in PBC has been reported [49]. The study was conducted from August 2017 to June 2021 across 14 centers in the UK. A total of 457 PBC patients with an inadequate response to UDCA were recruited. Overall, 259 patients received OCA and 80 received fibrates (fibric acid derivatives) and completed 12 months of therapy, yielding a dropout rate of 25.7% and 25.9%, respectively. Treatment efficacy was quantified by the proportion of patients attaining a biochemical response according to propensity score matching. The 12-month biochemical response rates were 70.6% with OCA and 80% under fibric acid treatment, without reaching any statistical significance.

With the objective of evaluating the time to first occurrence of liver transplant or death, OCA-treated patients in the POISE trial and open-label extension were compared with non-OCA-treated external controls [50]. Propensity scores were generated for external control patients meeting POISE eligibility criteria from 1381 patients in the Global PBC registry study and 2135 in the UK PBC registry. Over the 6-year follow-up, patients treated with OCA had a significantly greater transplant-free survival than comparable external control patients.

## 5. Combined Therapy with OCA and Fibrates

Fibrates, well-known agents with anti-lipidemic properties, were proposed as a second-line treatment because their beneficial effects on inflammation, cholestasis, and fibrosis are documented, resulting from their activity as peroxisome proliferator-activated receptor (PPAR) agonists. Fibrates have different affinities to the three main PPAR isoforms, PPARα, PPARβ/δ, and PPARγ, and consequently can activate different signaling pathways. As an example, fenofibrate, a PPARα agonist, upon binding to its receptor, increases the expression of multidrug resistance protein 3 (MDR3) [51]. Furthermore, it increases biliary phosphatidylcholine secretion, thus ameliorating a recognized biomarker of cholestasis. Bezafibrate acts as a dual agonist of PPARα and PPARγ and is also a pregnane X receptor (PXR) agonist [52]. The BEZURSO trial is a Phase III study, employing bezafibrate in combination with UDCA, and was the first placebo-controlled trial evaluating the use of fibrates as a second-line treatment for PBC. In this study, the second-line combination therapy of bezafibrate and UDCA was effective in obtaining a complete biochemical response with a rate significantly higher than that observed in patients treated with a placebo and UDCA [53]. This regression was associated with a concurrent improvement of both symptoms and surrogate markers of liver fibrosis. The most frequently reported ADRs of fibrates include increased levels of creatinine and transaminases and heartburn. As a consequence of its main mechanism of action involving a reduction in BA synthesis, clofibrate treatment can lead to the formation of gallstones and hypercholesterolemia [54], two events which have not been observed during treatment with fenofibrate or bezafibrate.

A triple therapy with UDCA, OCA, and fibrates was studied in a multicenter retrospective cohort of patients with PBC [55]. Fifty-eight patients were treated with a combination of UDCA (13–15 mg/day), OCA (5–10 mg/day), and fibrates (fenofibrate 200 mg/day or bezafibrate 400 mg/day). This combination achieved a significant reduction in ALP level compared to dual therapy (odds ratio for ALP normalization of 5.5). The primary outcome (change in ALP) and the effect on pruritus are summarized in Table 4.

## 6. Conclusions

In May 2021, the Food and Drug Administration issued a new warning restricting the use of OCA in patients with advanced cirrhosis (https://www.fda.gov/drugs/drug-safety-and-availability/due-risk-serious-liver-injury-fda-restricts-use-ocaliva-obeticholic-acid-primary-biliary-cholangitis, accessed on 1 September 2022). Advanced cirrhosis was defined on the basis of current or prior evidence of liver decompensation (e.g., encephalopathy, coagulopathy) or portal hypertension (e.g., ascites, gastroesophageal varices, or persistent thrombocytopenia). A practical guidance statement was published thereafter by the AASLD [56]. In this statement, the AASLD reported the contraindication on cirrhosis announced by the FDA, namely decompensated cirrhosis, and further recommended the careful monitoring of any patient with cirrhosis, even if not advanced, receiving OCA. In eligible patients, the recommended starting dose of OCA is 5 mg, which can be titrated to 10 mg after 6 months if OCA is well-tolerated. It is also recommended by the AASLD to monitor liver function before and after the initiation of OCA therapy.

In conclusion, due to its complex and fascinating mechanism, OCA represents a complete intervention for the therapeutic management of those PBC patients who cannot be treated satisfactorily with UDCA for efficacy or safety reasons. However, more real-world data are needed to gain a full understanding of its pharmacological and toxicological features.

## Figures and Tables

**Figure 1 biomedicines-10-02464-f001:**
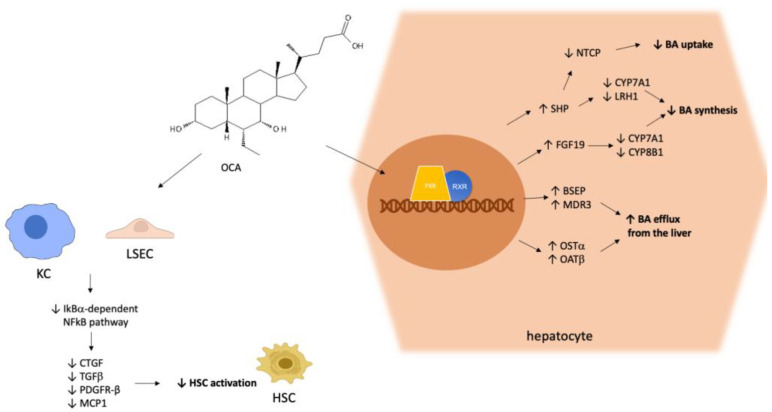
Molecular mechanism of hepatic OCA pharmacodynamics. OCA activates FXR, thereby triggering cellular pathways leading to a reduction in the synthesis and hepatic uptake of BAs, and an increase in their efflux from the liver. Furthermore, OCA acts on LSEC and KC, exerting anti-inflammatory and antifibrotic effects by reducing the production of proinflammatory cytokines and HSC activation, respectively. Abbreviations: farnesoid X receptor (FXR), retinoid X receptor (RXR), bile acid (BA), Kupffer cell (KC), liver sinusoidal endothelial cell (LSEC), hepatic stellate cell (HSC), small heterodimer partner (SHP), liver receptor homolog 1 (LRH-1), fibroblast growth factor-19 (FGF-19), sodium taurocholate co-transporting polypeptide (NTCP), bile salt export pump (BSEP), multidrug resistance protein-3 (MDR3), organic solute transporters (OST), transforming growth-factor β (TGFβ), connective tissue growth factor (CTGF), platelet-derived growth factor β-receptor (PDGFR-β), monocyte chemo-attractant protein-1 (MCP1), nuclear factor kappa-light-chain-enhancer of activated B cells (NF-κB), inhibitor of kB (IκB).

**Table 1 biomedicines-10-02464-t001:** Summary of the main pre-registration studies described in the text.

NCT Number [Ref]	Type of Study	Therapeutic Scheme	Population	Outcome	Adverse Events
NCT00570765 [31]	Phase II study, 3-month randomized, double-blind, placebo-controlled, parallel group phase, followed by a long-term safety extension (LTSE)	OCA monotherapy (10 or 50 mg)	60 PBC patients (18–70 years)	ALP reduction at both dosages after a 3-month treatment. Improvement of GGT, ALT, conjugated bilirubin, IgG	Pruritus (placebo 35%, OCA 10 70%, 94% OCA 50
NCT01473524 [32,33]	Phase III study, international 12-month randomized, double-blind (DB), placebo-controlled, parallel group phase, followed by a long-term safety extension (LTSE) phase of up to 5 years	OCA 5 mg (6 months) up to 10 mg or 10 mg vs. placebo	217 patients (≥18 years)	ALP reduction only after 12-month treatment with combination Reduction in total and direct bilirubin	Pruritus (56% in the 5–10% group and 68% in the 10 mg group vs. 38% placebo
NCT03253276 [38]	Early phase I, double-blind placebo-controlled crossover study	OCA vs. placebo	8 PBC patients	OCA reduced the time hepatocytes are exposed to potentially cytotoxic bile acids.	1 patient dropped for pruritus
NCT00550862 [39]	Phase II, randomized, double-blind study	OCA (10, 25, 50 mg) plus UDCA combination	165 patients (18–75 years)	Significant reduction in ALP, γ-GT, and ALT compared with placebo, in patients with PBC experiencing an inadequate response to UDCA	13% discontinuation for pruritus

**Table 2 biomedicines-10-02464-t002:** Ongoing clinical trials recruiting patients for post-registration efficacy assessment.

NCT Number	Type of Study	Therapeutic Scheme	Estimated Enrollment	Primary Endpoints
NCT05450887	Randomized, double-blind, multicenter, placebo-controlled phase III clinical trial	OCA (5 mg titrated to 10 mg) ± UDCA vs. placebo ± UDCA (13~15 mg/kg/day)	156 PBC patients (18–75 years)	Percentage of PBC patients reaching ALP < 1.67× Upper Limit of Normal, and ALP decrease ≥ 15% from baseline, and total bilirubin ≤ ULN after 12-month treatment
NCT03703076	Post-authorization non-interventional observational, multi-site study	OCA (5 or 10 mg)	150 patients	Response to Ocaliva^®^ after 12-month treatment (monotherapy or combination) assessed by Paris II response criteria
NCT05293938	Retrospective study	OCA (5 or 10 mg) and UDCA	2544 participants	Time to the first occurrence of the composite endpoint of all-cause death, liver transplant, or hospitalization for hepatic decompensation after 67 months
NCT05292872 (HEROES PBC)	Retrospective study	OCA (5 or 10 mg) and UDCA	3156 participants	Time to the first occurrence of all-cause death, liver transplant, or hospitalization for hepatic decompensation after 67 months
NCT05239468	Phase IIa, double-blind, randomized, active-controlled, parallel group study	Bezafibrate 100 or 200 mg, OCA 5 mg, Bezafibrate placebo, OCA placebo	60 patients	ALP change after 12 weeks vs. baseline
NCT04594694	Phase II, double-blind, randomized, parallel group study	Bezafibrate 200 or 400 mg, OCA mg, Bezafibrate placebos, OCA placebo	75 patients	ALP change after 12 weeks vs. baseline
NCT04076527	Prospective, multicenter cohort study	OCA vs. UDCA	1200 patients	Construction of a systematic registry to describe the characteristics and the recent state of usual clinical care of the respective population
NCT04956328	Multicenter, randomized, double-blind trial	OCA (5 to 10 mg) + UDCA, or placebo + UDCA	120 patients	Percentage of PBC patients reaching ALP < 1.67× ULN, and ALP decrease ≥ 15% from baseline, and total bilirubin ≤ ULN after 48week-treatment

**Table 3 biomedicines-10-02464-t003:** Real-world data in three cohorts of patients with PBC.

Author	Country	N. of Patients	Inclusion Criteria	% of Cirrhosis	% of pts with AIH/PBC Overlap	% of Response According to POISE
Roberts	Canada	64	Hepatologist’s discretion	23.7	6.3	18
Gomez	Spain/Portugal	120	Lack of response to Paris II criteria	21.7	10	29.5
D’Amato	Italy	191	ALP > 1.5 UNL	32	15	51.9

**Table 4 biomedicines-10-02464-t004:** Outcomes of triple therapy (UDCA + fibrates + OCA) [55].

Outcome	Baseline Dual	Baseline Triple	Last Follow-Up Triple
ALP (xULN)	2.5	1.8	1.1
Normal ALP (%)	0.7	10.3	47.4
Absence of pruritus	41.1	51.8	66.1

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
