# Peer review of "Obeticholic Acid for Primary Biliary Cholangitis"

_biomedicines, 2022, doi:10.3390/biomedicines10102464_

Round 1

Reviewer 1 Report

The authors summarize the pharmacologic actions and efficacy of obeticholic acid (OCA) in patients with PBC based on a number of reports. Although this is a clinically important study, some statements seem superficial. Therefore, several points need to be clarified and discussed.

1.    The authors should add the adverse reactions caused by OCA, such as pruritus, to the table.

2.    As you wrote in lines 165-166, liver stiffness increased in the OCA 5-10 mg group. Is there any other literature related to high/low-dose OCA? The relationship between the dose of OCA and the therapeutic effect should be discussed to verify whether low-dose OCA has no beneficial effect.

3.    The authors should make a table regarding the combination therapy with OCA and fibrates to highlight the improvement in combination therapy compared to monotherapy.

4.    We believe that this needs to be discussed, as there are differences in response rates reported in Canada, Spain/Portugal, and Italy.

5.    It would be better to add a description of the characteristics of UDCA refractory cases (Cheung AC, et al. Clinical Gastroenterology and Hepatology 2019;17:2076-2084. e2072).

6.    The authors should include citations for the Paris II criteria and Toronto criteria.

7.    The parentheses in line 269 and 272 are missing.

Author Response

The authors summarize the pharmacologic actions and efficacy of obeticholic acid (OCA) in patients with PBC based on a number of reports. Although this is a clinically important study, some statements seem superficial. Therefore, several points need to be clarified and discussed.

  1. The authors should add the adverse reactions caused by OCA, such as pruritus, to the table. 

We thank the Reviewer for this suggestion and modified the table accordingly.

  1. As you wrote in lines 165-166, liver stiffness increased in the OCA 5-10 mg group. Is there any other literature related to high/low-dose OCA? The relationship between the dose of OCA and the therapeutic effect should be discussed to verify whether low-dose OCA has no beneficial effect. 

We thank the reviewer for this useful comment. These issues have been added to the text.

  1. The authors should make a table regarding the combination therapy with OCA and fibrates to highlight the improvement in combination therapy compared to monotherapy. 

Table 4 has been added.

  1. We believe that this needs to be discussed, as there are differences in response rates reported in Canada, Spain/Portugal, and Italy. 

We thank the Reviewer for this comment and added a short discussion about this topic.

  1. It would be better to add a description of the characteristics of UDCA refractory cases (Cheung AC, et al. Clinical Gastroenterology and Hepatology 2019;17:2076-2084. e2072). 

We thank the Reviewer for this suggestion and modified the text accordingly.

  1. The authors should include citations for the Paris II criteria and Toronto criteria.

We thank the Reviewer for this suggestion and added the required citations.

  1. The parentheses in line 269 and 272 are missing.

We apologize for our mistake and fixed the text.

Reviewer 2 Report

Floreani et. al. conducted a narrative review addressing the updated available information in regards of the use of OCA as second-line therapy for UDCA non-responders in PBC patients.

Minor points

line 12 a connecting word is missing right after UDCA

line 23-24 should read progress not progresses 

line 26 replace the word commonest

line 36 positivity should be replaced by high titre 

line 198 the word respectively is missing

line 221 the sentence : After 12 months, 35%  of patients treated with OCA had a 35% of response according to POISE criteria...sounds confusing. re write.

Major points

line 73 it seems like figure 1 is not properly described. Authors  stated abbreviations only. A concise description of the scheme in the figure legend is recommended.

In this reviewer´s opinion a clearer conclusion is missing. Based on the manuscript OCA use is recommended when cirrhosis is not apparent yet.

Author Response

Floreani et. al. conducted a narrative review addressing the updated available information in regards of the use of OCA as second-line therapy for UDCA non-responders in PBC patients.

Minor points

line 12 a connecting word is missing right after UDCA

line 23-24 should read progress not progresses 

line 26 replace the word commonest

line 36 positivity should be replaced by high titre 

line 198 the word respectively is missing

line 221 the sentence : After 12 months, 35%  of patients treated with OCA had a 35% of response according to POISE criteria...sounds confusing. re write.

We thank the Reviewer for the careful revision and fixed all the minor issues listed above.

Major points

line 73 it seems like figure 1 is not properly described. Authors  stated abbreviations only. A concise description of the scheme in the figure legend is recommended.

We thank the Reviewer for this suggestion and modified the legend accordingly.

In this reviewer´s opinion a clearer conclusion is missing. Based on the manuscript OCA use is recommended when cirrhosis is not apparent yet.

We thank the Reviewer for this suggestion and modified the conclusion accordingly.

Round 2

Reviewer 1 Report

The authors properly responded to my queries. 

Reviewer 2 Report

Authors have substantially  improved the quality of the manuscript.